# The Role of Histology Alongside Clinical and Endoscopic Evaluation in the Management of IBD—A Narrative Review

**DOI:** 10.3390/jcm14072485

**Published:** 2025-04-05

**Authors:** Dorottya Angyal, Fruzsina Balogh, Talat Bessissow, Panu Wetwittayakhlang, Akos Ilias, Lorant Gonczi, Peter L. Lakatos

**Affiliations:** 1Department of Internal Medicine and Oncology, Semmelweis University, 1083 Budapest, Hungary; adorka99@gmail.com (D.A.);; 2Division of Gastroenterology, Central Hospital of Northern Pest, Military Hospital, 1062 Budapest, Hungary; 3Division of Gastroenterology, McGill University, Montreal, QC H3A 0G4, Canada; 4Gastroenterology and Hepatology Unit, Division of Internal Medicine, Faculty of Medicine, Prince of Songkla University, Hat Yai 90110, Thailand; wet.panu@gmail.com

**Keywords:** inflammatory bowel disease, ulcerative colitis, Crohn’s disease, histological remission, disease monitoring, treat-to-target

## Abstract

Inflammatory bowel diseases (IBD), including Crohn’s disease (CD) and ulcerative colitis (UC), are chronic inflammatory conditions requiring continuous monitoring. Today, endoscopy is the gold standard for assessing disease activity, with histological evaluation providing additional insights. Studies suggest that persistent histological inflammation, despite endoscopic remission, may be associated with a higher risk of relapse in UC, suggesting its role in treatment decisions. In CD, histological assessment is limited by its patchy nature, transmural inflammation and lack of validated scoring systems. Few retrospective studies with conflicting results have examined the prognostic value of histological remission in CD, and its role in predicting long-term outcomes remains unclear. This narrative review aims to summarize and discuss the available evidence regarding the additional value of histological assessment in IBD management. In UC, the ongoing VERDICT study is expected to provide evidence on the impact of incorporating histological remission as a treatment target compared to a strategy based on clinical and endoscopic activity. Recently published interim results indicate that targeting histological remission does not lead to better clinical/biochemical disease activity. Thus, while patients achieving histological healing are associated with better outcomes, the question arises whether achieving histological remission is an intrinsic (biological) characteristic of the patient and indicator of an easier to treat patient group or a result of more effective therapy.

## 1. Introduction

Inflammatory bowel diseases (IBD), such as Crohn’s disease (CD) and ulcerative colitis (UC), are chronic, idiopathic inflammatory disorders of the gastrointestinal tract, significantly affecting patients’ quality of life and leading to complications and disability. Evolving therapeutics allow physicians to choose between different treatment options, therefore it is essential to optimize the management of the disease to improve long-term outcomes.

Conventionally the therapy of IBD was based on the control of symptoms. However, there is clear discordance between the presence and severity of symptoms and mucosal status on endoscopic assessment, especially in patients with CD [1,2]. There is accumulating evidence that treatment strategies targeting symptomatic remission fail to change the natural course of IBD [3,4,5]. The prospective randomized CALM study showed that the adjustment of therapy based on close monitoring of clinical symptoms combined with inflammatory biomarkers leads to better long-term endoscopic and clinical outcomes than symptom-driven decisions alone [6].

In the last decade therapeutic targets shifted from approaching the control of symptoms to objective disease monitoring aiming to alter the disease course and preventing long-term disease complications. Newer management strategies focus also on the control of inflammation including endoscopic and histological healing and normalization of inflammatory biomarkers such as CRP and fecal calprotectin (FC) [5].

In 2015 the Selecting Therapeutic Targets in Inflammatory Bowel Disease (STRIDE) consensus introduced the treat-to-target (T2T) approach which aims to achieve remission by adjusting therapy based on the fulfillment of specific treatment targets [7]. The original consensus was revised in 2021 in STRIDE-II, encompassing evidence- and consensus-based recommendations for T2T strategies in IBD [8].

In STRIDE-II, clinical remission and normalization of CRP and FC were identified as short medium-term treatment targets whereas endoscopic healing was named as a long-term target in both CD and UC. Transmural healing and histological remission were not considered as treatment targets in either CD or UC. Nonetheless, transmural healing in CD and histological remission in UC could be used as an adjunct to endoscopic remission to represent a deeper level of healing [8]. In contrast, while additional noninvasive biomarkers (serological markers) can help in better characterization of the disease phenotype, they do not have a role in the assessment of disease activity [9].

Today, endoscopic evaluation with ileocolonoscopy is the gold standard in the assessment of disease activity. In fact, it was always part of the clinical scores in patients with UC (e.g., Mayo score). Mucosal healing on endoscopy is associated with lower rates of relapses, hospitalizations, surgery and/or colorectal neoplasia [10,11,12,13,14,15,16,17].

However, in many patients, histological assessment may show persistent microscopic inflammation despite endoscopic healing [18,19,20]. Thus, there is an ongoing debate regarding the role of histological evaluation in clinical decision making, especially in cases with mild or no endoscopic activity. Disease “clearance” and “treat-to-clear” became fashionable terms in recent years, defining a state of deep remission with the completion of symptomatic, endoscopic and histological remission [21,22]. The main question is whether therapy adjustments aiming for histological remission are beneficial considering long-term outcomes compared to patients with clinical and endoscopic remission.

This narrative review aims to summarize and discuss the currently available evidence regarding the additional value of histological assessment in the management of UC and CD and its use in everyday clinical practice.

## 2. Histological Scoring Systems

There are several histological scoring systems available for the assessment of disease activity in UC. However, only the Robarts histopathology index (RHI) and the Nancy histopathology index (NHI) have been fully validated, while the Geboes score (GS), although partially validated, is the most widely used. The European Crohn’s Colitis Organisation (ECCO) recommended the use of RHI and NHI in randomized controlled trials and NHI in observational studies or in clinical practice [23]. The main histological features assessed are similar in all three scoring indices, including chronic inflammation, neutrophilic inflammation and surface epithelial injury.

An observational cohort study comparing the GS, RHI and NHI found considerably high concordance concerning the distinction between patients in histological remission or activity, especially when comparing the NHI and RHI. However, the differences in the number of characterized stages and the definition of inflammatory cell infiltrate can cause discrepancies [24]. A standardized definition of histological remission is still lacking, remission criteria in mentioned scoring indices are summarized in Table 1 [23,25,26,27,28,29].

In contrast, in CD there is no fully validated mucosal histological disease activity score. The main concern regarding histological assessment in CD is the risk of sampling error due to the discontinuous and patchy nature of the disease and the transmural inflammation which may not be represented on mucosal biopsies. The Global Histological Activity Score (GHAS) is the most used index in CD which also lacks formal validation. The ECCO recommends the use of NHI for the histological evaluation of CD biopsies in clinical practice. However, it does not provide a complete definition of histological healing and normalization in CD [30].

Further limitations of histological evaluation are the significant intra- and interobserver variability, biopsy localization difficulties, the invasive nature of the assessment and the cost in terms of money and time. In addition, the cost-effectiveness of histological assessment is unproven, as well as there being a clear difficulty of appropriate sampling/lack or reach in patients with small bowel CD.

## 3. Histological Endpoints in Clinical Trials

Recently histological assessment became an accepted endpoint in clinical trials in patients with UC. The Food and Drug Administration’s (FDA) 2016 industrial guideline named histological evaluation as an endpoint to support claims of mucosal healing in UC in phase 2 trials aiming to effectively incorporate histological assessments in phase 3 trials [31]. Soon thereafter, in 2018, the European Medicines Agency (EMA) included histological evaluation of mucosal inflammation as a secondary endpoint in the clinical development of new medicinal products in UC [32]. Moreover, in 2020 an ECCO position paper declared histological remission as a treatment target in UC while expressing the need for standardization of histological procedures, definitions and scoring systems [23].

However, in 2022 the new FDA guideline did not recommend the use of the term “mucosal healing” due to a lack of consensus on definition and listed histological response/remission only as exploratory endpoints [33].

## 4. Histologic Remission and Clinical Outcomes in UC

Data available from retrospective and few prospective observational cohort studies show that achieving histological remission in UC results in better clinical outcomes compared to clinical and endoscopic remission. A meta-analysis by Park et al. [34] in 2016 reported that approximately 30% of patients with clinical and endoscopic remission had persistent histological activity. Histological healing was superior to endoscopic and clinical remission in predicting clinical outcomes. There was a 52% relative risk reduction in the probability of clinical relapse/exacerbation in patients with histological healing compared with patients without [34]. However, a limitation of Park’s meta-analysis is that it included all studies where patients were in clinical remission, irrespective of the presence of endoscopic activity at baseline. This may introduce a significant bias since there is a strong correlation between histological and endoscopic activity, so the net effect of histological activity cannot be objectively assessed, and the importance of histological healing can be overestimated. The main question is whether there is an independent effect of histological activity on long-term outcomes and whether this can be best evaluated in patients in endoscopic remission with or without persistent histological activity.

Gupta et al. published a meta-analysis examining the association between histological activity and relapse in patients with CD or UC in endoscopic remission [35]. A systematic search of literature from inception to June 2020 was conducted including 27 studies comprising 2677 UC patients with a Mayo endoscopic subscore (MES) of 0 or 1 assessing the risk of relapse with or without histological disease activity. Definition of relapse was heterogeneous in the published papers, most studies used the criteria of clinical relapse while some required therapy change or also included colectomy and hospitalization. In their analysis, histologically active disease was associated with an increase in risk of relapse (OR, 2.41; 95% CI, 1.91–3.04). The impact of histological activity on relapse was similar in the studies that included patients with a MES of either 0 or 1 (OR, 2.34; 95% CI, 1.80–3.05) when compared with studies that provided data only on patients with a MES of 0 (OR, 2.66; 95% CI, 1.54–4.58) (*p* = 0.68), while the impact was greater in studies that included at least 24 months of follow-up (OR, 3.49; 95% CI, 2.53–4.82).

In another meta-analysis from 2020, Yoon et al. examined the impact of histological activity on relapse in UC patients achieving the composite endpoint of clinical remission and MES 0 or 1 and performed a subanalysis in patients with a MES of 0 [36]. They found histological remission to result in a 61% risk reduction in clinical relapse in the case of clinical remission and a MES of 0 or 1 (RR, 0.39 (95% CI, 0.31–0.51)) relative to persistent histological activity. Their subanalysis in patients achieving a MES of 0 reported a similar, 63% risk reduction in clinical relapse (RR, 0.37 (95% CI, 0.24–0.56)) which translates to a 5% (95% CI, 3.3–7.7) estimated annual clinical relapse risk in case of histological remission compared to 13.7% in patients with persistent histological activity.

Both Gupta and Yoon observed a stronger correlation of histological activity and risk of relapse in studies that defined histological remission using validated histological indices [35,36]. Gupta et al. also performed a subgroup analysis by various definitions of histological activity and scoring systems. A cutoff of GS < 3.1 demonstrated a significant association between histological activity and relapse, whereas more rigorous GS cutoffs (<2.1 or 0) demonstrated a stronger effect size with numerically greater impact on relapse rates. A NHI > 2 for histological activity also demonstrated higher rates of relapse. When evaluating individual components of histological activity, basal plasmacytosis, neutrophilic infiltrations, mucin depletion, and crypt architectural irregularities all demonstrated association with higher rates of relapse, while neither crypt abscesses nor chronic inflammatory infiltrate were predictive of relapse [35].

To further evaluate the published data, we collected and reviewed cohort studies assessing the impact of histological activity on risk of relapse in adult UC patients in endoscopic remission (MES 0 or 1) published since June 2020 [37,38,39,40,41,42,43,44,45] (Table 2).

In general, outcomes were in line with the earlier findings; however, outcomes such as the need for hospitalization or colectomy were not separately assessed in most studies, presumably due to the low number of patients. Wei et al. provided data on hospital admission times (0.33 ± 0.90 and 0.47 ± 1.17, *p* = 0.647) and emergency department visit times (0.04 ± 0.21 and 0.42 ± 1.22, *p* = 0.197) in histologically inactive and active groups, respectively. They found numerically lower rates of hospitalization in case of histological remission without statistical significance [45].

Seong et al. [39] reported significant association between histological activity and clinical relapse in the total cohort with the inclusion criteria of MES 0 or 1, but no such association was found in the subgroup of patients with a MES of 1 at baseline. In addition, in the subgroup of patients with histologic improvement, there was no relevant difference in the risk of clinical relapse between the patients with MES 0 and those with MES 1; also, endoscopic activity was not proven to be a significant risk factor in multivariable analysis.

Currently, data is only available from observational cohort studies. The first randomized controlled trial (RCT) providing evidence on the role of histological remission as a treatment target in UC will be the ongoing actiVE ulcerative colitis, a RanDomIsed Controlled Trial (VERDICT) [45]. The CALM study was the first RCT providing evidence on the superiority of CD management based on close monitoring of clinical symptoms combined with biomarkers over symptom-based decision-making, resulting in better clinical and endoscopic outcomes [6].

The VERDICT is expected to be a similar milestone in UC, comparing the outcomes of symptom vs. symptom+endoscopy vs. symptom+endoscopy+histology-based objective treatment targets. This study will also shed light on the relative importance of histological healing. An interim analysis was presented in 2025 and the final results are expected in 2026. The aim is to include 660 patients with moderate to severe UC and treat them to the following treatment targets: corticosteroid free (CSF) clinical remission (group 1), or clinical+endoscopic remission (group 2) or clinical+endoscopic+histological remission (group 3) using vedolizumab therapy. Patients will be assessed every 16 weeks and therapy will be intensified if the target is not reached, and patients are followed up to 80 weeks after reaching the target. The primary endpoint is time from treatment target achievement to a UC-related complication [46].

Interim data of the VERDICT were very recently reported at the 2025 ECCO congress on biomarker changes (FC and CRP) from baseline to weeks 8, 16, 32 and 48 [47]. The proportion of patients not achieving the treatment target at week 48 were 9.7%, 22.7% and 32.2% in groups 1, 2 and 3, respectively. The proportion of patients not achieving the treatment target in the bioexposed patients in group 3 was as high as 46.2% Biomarker levels improved from baseline to Week 16, with incremental improvements through to Week 48. Of note, the proportion of patients achieving the target of CSF biomarker remission (FC < 250 mg/kg, CRP < 5 mg/L) was not higher in group 3 at week 32 and 48. Interestingly, the proportion of biomarker remission over time in the 3 groups was similar through weeks 8 to 48, with no advantage for group 3. The biomarker remission rates at week 48 were 51.6%, 57.1% and 48.6% in groups 1, 2 and 3.

Finally, CSF histological remission in group 3, with 265 included patients, was also reported [48]. At week 16, 50% of patients were in histologic remission, with 66.7% at week 48, with similar assessments for RHI, NHI and GS. Histological healing rates were not reported so far from groups 1 and 2.

A recently presented prospective cohort study of 253 UC patients in clinical remission at baseline with 3- and 5-year follow-up found that histological activity was not predictive of long-term clinical relapse. In contrast, endoscopic activity of MES 0 compared to MES 1–3 at baseline predicted clinical relapse at 5 years (*p* = 0.0080) and IBD related ER visit at 3 years (*p* = 0.02). These findings question the additional role of histology in the management of UC in complete endoscopic remission [49].

## 5. The Role of Histological Healing in Crohn’s Disease

In contrast to UC, the importance of histologic healing/remission on outcomes in CD patients with endoscopic remission is less clear. There are many possible confounders that make objective assessment difficult: the patchy and transmural nature of inflammation and differences in location (small vs. large bowel involvement), leading to possible sampling errors. It is also not clear if the UC histology scores are fully appropriate to assess the histological inflammation in CD patients.

Few retrospective studies are available, and they show conflicting results [35,50,51,52,53,54]. Yoon et al. [54] reported 43% lower risk of treatment failure (clinical flare requiring treatment modification, hospitalization and/or surgery) in patients who achieved histologic remission over a 2-year follow-up. This equals a 12.9% vs. 18.2% 1-year cumulative risk of treatment failure in case of histologic remission vs. persistent histologic activity in a retrospective cohort study of 215 CD patients in clinical and endoscopic remission published in 2021. Christensen et al. [51] performed a retrospective study of 101 patients with CD limited to the terminal ileum in clinical remission at baseline investigating the impact of endoscopic and histological healing on outcomes. It was found that histologic healing, but not endoscopic healing, is associated with decreased risk of clinical relapse, (HR, 2.05; 95% CI, 1.07–3.94; *p* = 0.031), medication escalation (HR, 2.17; 95% CI, 1.2–3.96; *p* = 0.011) and corticosteroid use (HR, 2.44; 95% CI, 1.17–5.09; *p* = 0.018). In contrast, Hu et al. [50] analyzed the data of 129 CD patients in endoscopic remission using an institutional registry and found no significant difference in clinical relapse rates (dose escalation, change in therapy, need for systemic steroids or CD-related hospitalization or surgery) based on ileal (*p* = 0.43) or colonic (*p* = 0.73) histological activity.

Finally, a recently published post hoc analysis of the prospective multicenter STORI cohort study evaluated the impact of histological remission for predicting clinical relapse in CD. Biopsies of 76 CD patients in clinical remission were assessed using multiple histological indices. While endoscopic remission was associated with a lower relapse rate, there was no significant difference in relapse rates based on the achievement of histological remission in the total cohort, nor in the subgroup of patients with endoscopic remission. On the other hand, they found a very high correlation between the NHI, RHI, GS and IBD-DCA scores in assessing histological inflammation in CD [55].

Thus, it is not clear if achieving histological healing provides a superior target in patients with CD. Results of the above-mentioned studies are summarized in Table 3.

## 6. Expert Opinion: Unanswered Questions/Future Directions

The data available so far confirm that histological healing can be achieved in a proportion of patients. It also confirms that patients achieving histological healing may be associated with better clinical outcomes (flare, hospitalization or also surgery).

However, at least with the current approach(es) a large proportion of patients cannot achieve histological healing and recent data from the VERDICT study [47] suggest that biomarker remission rates are similar after 1 year if we target to achieve clinical remission, mucosal healing or histological healing. Thus, at least in the above study design, there is a plateau of reaching biomarker (and maybe histological) remission. Of note, comparative histological healing rates were not reported. This would mean that patients with histological healing are biologically different, i.e., “they are able to completely heal”, and this is why the outcomes thereafter are better, not due to the more aggressive treatment approach. The VERDICT study [46] will provide invaluable data, since comparative healing rates will be available for the 3 groups with different treatment targets, and if authors will compare flare/failure rates in the 3 groups in patients achieving histological remission, the above question will be answered.

Another major aspect that has not been thoroughly discussed is whether therapy change is cost-effective in the case of histological activity despite endoscopic and clinical remission considering the relatively small absolute decrease in annual clinical relapse risk (app. 5% vs. 13.7%). Dose optimization/increase is clearly associated with additional costs and possible additional side-effects, whereas evidence demonstrating the benefit of treatment escalation pursuing a state of deeper remission is lacking. Again, it is not clear if the achievement of histological remission is characteristic of the patients with different (“milder”) disease or whether it is the result of a more effective treatment. Furthermore, histological remission may be achieved by maintaining the ongoing therapy that resulted in endoscopic remission or by a change in therapy. More clear answers to the former questions are expected with the final results of the VERDICT study [46].

A further important aspect/limitation in clinical practice is the known association between IBD and irritable bowel syndrome (IBS). According to a recent metaanalysis [56] IBS could be present in up to 25% of patients with histological remission, and this can generate additional costs due to repeated outpatient emergency room visits, reassessment and even hospitalizations; thus, it is questionable if and to what extent the possible beneficial effects of histological healing would be associated with improved outcomes and cost savings in real-world cohorts.

While there are accumulating data on the impact of histological activity on long-term outcomes in asymptomatic UC patients, the prognostic value of histological features in active UC has not been much studied. Zhao et al. conducted a prospective cohort study of biological-naïve patients with UC, which enrolled 150 patients undergoing ileocolonoscopy with sampling of mucosal biopsies within 3 months prior to biological therapy initiation [57]. In the later study, the higher initial histological disease activity measured by NHI or GS was found to be an independent predictor of colectomy following biological treatment, while endoscopic disease activity was not. Furthermore, a lower rate of clinical response was reported in patients with severe histological disease at baseline. These findings support the assumption that severe/persisting histological activity is rather an indication of a more severe phenotype of UC than inadequate medical therapy. Of note, the importance of histology was not studied in UC patients with ASUC (acute severe ulcerative colitis); thus, conclusions on its importance cannot be extrapolated to this patient group.

Finally, recent data challenge the hypothesis of continuous inflammation/healing in UC and suggests that histological healing rates may vary across the colonic segments, not related to applying topical therapy. In a recent prospective cohort study of 203 UC patients by the McGill group, poor correlation was found between the histological activity of mucosal biopsies taken from the rectosigmoid colon and other colon segments in clinical remission, challenging the role of flexible sigmoidoscopy in disease activity evaluation in UC [58]. In addition to the above, different healing in the rectum and in other segments of the colon is a well-known phenomenon and may occur in patients receiving a combination of oral/intravenous and/or topical therapy.

On the other hand, the concept of treating histological healing is much less established in CD. Confounders include the patchy and transmural nature of inflammation, possible sampling errors, difference of importance and probability of healing in the small bowel, terminal ileum or colon. Finally, data are simply lacking.

Whether the use of artificial intelligence [59] in the assessment of histological samples can overcome some of the above limitations and can lead to more accurate prediction of outcomes needs to be tested in future studies.

Finally, we would like to emphasize again that histological assessment is invasive, and an endoscopic evaluation is also needed for the sampling. Thus, the cost-effectiveness, safety and acceptance by the patients of additional invasive testing has to be proven over the clinical biomarker assessment.

Lastly, we provide a practical guide (Figure 1) for monitoring IBD patients according to treatment goals in line with the ECCO colorectal surveillance guideline [60].

## 7. Conclusions

There is increasing evidence for the benefit of histological healing in patients with ulcerative colitis. Patients achieving histological healing were found to have improved clinical and long-term outcomes. In contrast, it is still questionable if histological healing is an indicator of a different/special patient subgroup or if more aggressive treatment (treating to target) may help to achieve histological healing in the majority of patients. Cost-effectiveness of treating to histological healing must be established and more data is needed in patients with CD.

## Figures and Tables

**Figure 1 jcm-14-02485-f001:**
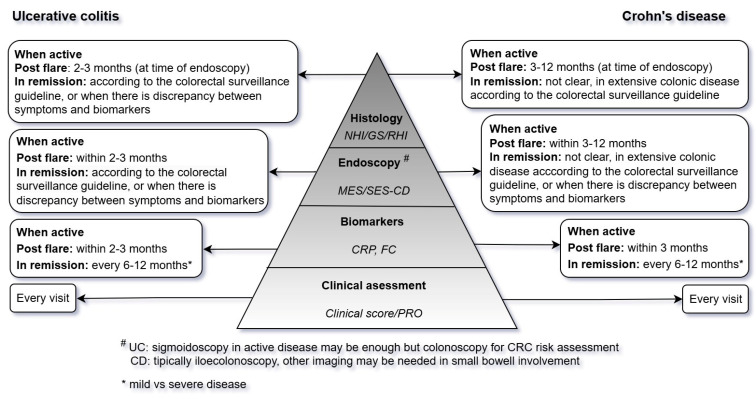
Practical guide for monitoring IBD patients according to treatment goals. NHI: Nancy histopathology index, GS: Geboes score, RHI: Robarts histopathology index, MES: Mayo endoscopic subscore, SES-CD: simplified endoscopic activity score for Crohn’s disease, CRP: C reactive protein, FC: fecal calprotectin, PRO: patient reported outcome, UC: ulcerative colitis, CD: Crohn’s disease, CRC: colorectal cancer, IBD: inflammatory bowel diseases.

**Table 1 jcm-14-02485-t001:** Summary of histological definitions and indices [23,25,26,27,28,29].

Target	Definition	GS	RHI	NHI
Histological response /improvement	Neutrophil infiltration in <5% of crypts, no crypt destruction, erosions, ulcerations or granulation tissue	≤3.0	≥7-point reduction or ≤9	≥1-point reduction or ≤1
Histological remission	Colorectal mucosa without neutrophilic inflammation ± indications of chronicity	≤2.0	≤3	0
Histological normalization	Complete normalization of the colorectal mucosa	0.0, 1.0, 2A.0, 2B.0, 3.0, 4.0 and 5.0	-	-

**Table 2 jcm-14-02485-t002:** Summary of cohort studies assessing the impact of histological remission on risk of relapse of patient with UC in endoscopic remission published since June of 2020 [37,38,39,40,41,42,43,44,45].

First Author	Year	Study Design	N *	Inclusion Criteria	Histological Cut-Off	Definition of Relapse /Disease Progression	*p*-Value
Shin [37]	2024	prospective	117	MES 0, 1	NHI ≥ 3	escalation or alteration of medication, surgical procedure, hospitalization for UC exacerbation	0.009
					NHI > 2		0.193
Wang [38]	2023	retrospective	74	MES 0, 1	NHI ≥ 2	pMayo > 2, MES > 1, start of steroids, hospitalization, escalation or alteration of therapy because of symptoms	0.002
Seong [39]	2023	retrospective	492	MES 0, 1	GS ≥ 3.1	changes in medication, hospitalization, colectomy and the development of colorectal cancer	<0.001
				MES 1			0.223
George [40]	2023	retrospective	445	MES 0, 1	RHI > 3	change in medical therapy, new steroid use, UC-related hospitalization and/or colectomy	0.008
Park [41]	2022	retrospective	142	UCEIS 0, 1	RHI > 3	escalation of therapeutic drugs, a visit to an emergency department and hospitalization	0.035
Jangi [42]	2021	retrospective	89	MES 0, 1	acute inflammation	recurrence of any rectal bleeding with an increase in stool frequency	<0.05
Narula [43]	2020	retrospective	269	MES 0	acute inflammation	symptoms suggestive of active UC disease activity requiring medical therapy escalation, hospitalization or colectomy	0.85
Kim [44]	2024	retrospective	435	MES 0, 1	GS ≥ 3.1	change or escalation of medication, hospitalization or total colectomy due to the aggravation of UC	0.03
Wei [45]	2024	retrospective	42	MES 0	NHI > 0	presence of clinical symptoms including abdominal pain, bloody stools, diarrhea and tenesmus, accompanied by endoscopic evidence of inflammation	0.006

* Number of patients.

**Table 3 jcm-14-02485-t003:** Summary of cohort studies assessing the impact of histological remission on risk of relapse of patient with CD. (clinical remission = CR, endoscopic remission = ER) [50,51,52,54,55].

First Author	Year	Study Design	N *	Inclusion Criteria	Histological Cut-Off	Definition of Relapse /Disease Progression	*p*-Value
Reenaers [55]	2024	prospective	76	CR	NHI > 0	CDAI > 250 or a CDAI increase of 70 points over two weeks	0.26
			45	ER			0.18
Yoon [54]	2021	retrospective	215	ER	NHI ≥ 2	composite of clinical flare requiring treatment modification, hospitalization or surgery	0.026
Hu [50]	2021	retrospective	129	ER	histologic activity	dose escalation, change in therapy, need for systemic steroids, hospitalization or surgery	0.73
Christensen [51]	2020	retrospective	105	CR **	histologic activity	HBI > 4 that resulted in alteration or addition of medical therapy, hospitalization or surgery	0.008
Brennan [52]	2017	retrospective	62	CR	histologic activity	increase in clinical disease activity requiring a change in medication	<0.05

* number of patients. ** CD limited to the terminal ileum was also inclusion criteria.

## Data Availability

Not applicable.

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
