# Peer review of "The Role of Histology Alongside Clinical and Endoscopic Evaluation in the Management of IBD—A Narrative Review"

_jcm, 2025, doi:10.3390/jcm14072485_

Round 1
Reviewer 1 Report
Comments and Suggestions for Authors
The review by Angyal D. et al. addresses the significance of histological remission in ulcerative
colitis and Crohn's disease, which is an active area of research and a highly debated topic.
The article is well-structured, with sections that define the concept of histological remission,
analyze its implications in clinical trials, and discuss evidence relevant to clinical practice.
Despite the interesting topic and the appropriate organization of the review, I have identified
several syntactic inaccuracies that would benefit from revision by a native speaker before
publication.
For instance, I recommend that the authors carefully revise the sentences between lines 194 and
205, as well as those between lines 227 and 230, to improve clarity and readability. Another
passage that, in my opinion, requires revision is the one referring to the 2016 meta-analysis by
Park and colleagues (lines 130–138). In citing this meta-analysis, the authors first mention the
percentage of patients with clinical and endoscopic remission but persistent histological activity,
and then state that the meta-analysis included only studies based on the requirement of clinical
remission, regardless of endoscopic activity. I suggest that the authors revise this passage,
possibly reordering the statements, to enhance clarity and ensure a more logical flow of ideas.
I find the study interesting and worthy of publication. However, substantial linguistic,
syntactic, and grammatical revisions are essential before it can be accepted.
Author Response
We thank the reviewer for finding our paper well-structured and the topic important. We thank for the detailed linguistic assessment and highlighting sections with syntactic inaccuracies. During the revision we carefully read and revised the sections highlighted to increase readability and make the text more straightforward
Reviewer 2 Report
Comments and Suggestions for Authors
I congratulate the authors on their paper entitled "The role of histology alongside clinical and endoscopic evaluation in the management of IBD – a narrative review". The paper provides a solid summary of histological assessment in IBD. There are - however - some minor issues with the paper that need to be addressed
- the CD section is a bit "underfilled" compared to the UC section, which is somewhat understandable given the fewer studies in that regards. Please add a table similar to the UC section detailing available studies
- add a limitations section to the use of histology in IBD follow-up. lines 109-111 allude to this alongside the expert opinion section, but a proper subsection is needed in my opinion that should be separate from the expert opinion/future directions. Items which you discuss there may the invasive nature of biopsies with the risks (albeit minor) associated to it, unclear cost-effectivness compared to endoscopic +/- laboratory work-up, the fact that small-bowel CD can rarely be biopsied alongside the interobserver variablilty associated with histopathological interpretation.
- a paragraph or some sentences on the risks associated with treating towards normal histology, especially with regards to the increased immunosuppression and malignancy, needs to be added. This is especially relevant for patients with normal CRP and calprotectin levels.
- add a flow-chart which integrates - in clinical practice - how histological assessment of disease activity could be used alongside other established methods (endoscopic assessment, laboratory work-up)
- there needs to be - somewhere in the text - an acknowledgement of the fact that normal CRP and calprotectin with the presence of symptoms may suggest functional "IBS-IBD" overlap. this is clearly demonstrated by multiple studies including this meta-analysis "https://doi.org/10.1016/S2468-1253(20)30300-9" where 25% of patients with histological remission still had symptoms. this would also be highly relevant to be added as an extra "arrow" in the flow-chart proposed above.
- given the hype surrounding Artificial intelligence and histological assessment in IBD (with papers including but not limited to https://doi.org/10.1016/j.dld.2024.05.033 AND https://doi.org/10.1016/j.modpat.2023.100124 AND review https://doi.org/10.1016/S2468-1253(24)00053-0 AND review https://doi.org/10.1177/17562848251325525 ), there needs to be a mention of that in the text with appropriate referencing
Author Response
I congratulate the authors on their paper entitled "The role of histology alongside clinical and endoscopic evaluation in the management of IBD – a narrative review". The paper provides a solid summary of histological assessment in IBD. There are - however - some minor issues with the paper that need to be addressed.
We thank the reviewer for the positive opinion and for finding our paper relevant and important.
- The CD section is a bit "underfilled" compared to the UC section, which is somewhat understandable given the fewer studies in that regards. Please add a table similar to the UC section detailing available studies.
Thank you for the comment, we added additional paper(s) and a Table summarizing the data in CD, however we should note that there or much less data on the importance of histology in CD
- Add a limitations section to the use of histology in IBD follow-up. lines 109-111 allude to this alongside the expert opinion section, but a proper subsection is needed in my opinion that should be separate from the expert opinion/future directions. Items which you discuss there may the invasive nature of biopsies with the risks (albeit minor) associated to it, unclear cost-effectivness compared to endoscopic +/- laboratory work-up, the fact that small-bowel CD can rarely be biopsied alongside the interobserver variablilty associated with histopathological interpretation.
Thank you for the comment, we modified this section and added the note in addition to invasiveness (which was already included) on lack of data on cost-effectiveness, and problems with sampling in the small bowel in CD patients.
- A paragraph or some sentences on the risks associated with treating towards normal histology, especially with regards to the increased immunosuppression and malignancy, needs to be added. This is especially relevant for patients with normal CRP and calprotectin levels.
Thank you for the comment, this is a controversial issue. In IBD patients improved control of inflammation is thought to be preventing malignancy (this inflammation-cancer cycle is well proven in patients with UC) and the current biologicals even in higher doses are not associated with malignancy risk with the exception of the controversial association between IFX and melanoma or IFX-immunosuppressive combination and lymphoma which is not the primary choice in the everyday practice in recent years. Therefore, we did not modify this section.
- Add a flow-chart which integrates - in clinical practice - how histological assessment of disease activity could be used alongside other established methods (endoscopic assessment, laboratory work-up).
Thank you for the comment, we added an additional Figure showing the flowchart, how histology can be implemented in the current management work-up scheme
- There needs to be - somewhere in the text - an acknowledgement of the fact that normal CRP and calprotectin with the presence of symptoms may suggest functional "IBS-IBD" overlap. This is clearly demonstrated by multiple studies including this meta-analysis "https://doi.org/10.1016/S2468-1253(20)30300-9" where 25% of patients with histological remission still had symptoms. This would also be highly relevant to be added as an extra "arrow" in the flow-chart proposed above.
Thank you, we completely agree with the comment and added this to the revised paper in the Expert opinion section
- Given the hype surrounding Artificial intelligence and histological assessment in IBD (with papers including but not limited to https://doi.org/10.1016/j.dld.2024.05.033 AND https://doi.org/10.1016/j.modpat.2023.100124 AND review https://doi.org/10.1016/S2468-1253(24)00053-0 AND review https://doi.org/10.1177/17562848251325525 ), there needs to be a mention of that in the text with appropriate referencing
Thank you, we added a short sentence on the possible present/future role for AI in the histological assessment in the Expert opinion section.
Reviewer 3 Report
Comments and Suggestions for Authors
This is a narrative review addressing the currently available evidence regarding the additional value of histological assessment in the management of ulcerative colitis (UC) and Crohn’s disease (CD) and its use in the clinical practice due to the ongoing debate regarding the role of histological evaluation in clinical decision making, particularly in mild or no endoscopic activity cases. The manuscript addresses a clinically relevant topic. However, the authors should expand the literature data on the accuracy of proposed non-invasive tools to follow and assess disease activity and treatment response and to avoid the invasive assessment by endoscopic and histological examinations.
The authors could stress the superiority of histological assessment recalling literature studies demonstrating that some serological markers, initially regarded as disease-specific such as Anti-saccharomyces cerevisiae antibodies (ASCA) for Crohn's disease that are however detectable also in a significant number of asymptomatic and symptomatic patients with celiac disease, as previously demonstrated (Aliment Pharmacol Ther. 2005 Apr 1;21(7):881-7. doi: 10.1111/j.1365-2036.2005.02417.x.; Gut. 2006 Feb;55(2):296.), and anti-neutrophil cytoplasmic (ANCA) that are also frequently detected in patients with autoimmune liver diseases such as autoimmune hepatitis and primary sclerosing cholangitis.This therefore suggest that there are no specific immuno-serological markers able to replace the histological evaluation. Therefore, these important literature data should be recalled to further highlight the added value of histological assessment in patient management.
Author Response
We thank the reviewer for finding the topic of our paper clinically relevant. We also agree that the non-invasive methods are important, this is what we already highlight in the introduction (STRIDE I and II criteria) starting with the clinical assessment (score/PROs) and continuing with CRP/FCAL.
The topic of the paper was to summarize the new data on the importance of histological assessment thus we did not intend to summarize the available data on biomarkers nor the serological markers (on which the authors have also performed important research earlier).
Therefore, following your comments, we added a short reminder on the importance and possible benefit(s) of less invasive assessment in the expert opinion section as well as added a comment on the possible use of serology for better patient characterization (we cited the landmark paper of our group, which was one of the early ones comprehensively assessing the importance of the serological markers in IBD ) and occasionally aid in diagnosis but clearly highlighted that they do not have a role in disease activity assessment (and this is the topic of the present paper).
Again, we would like to thank the reviewer for the thoughtful comment.
Round 2
Reviewer 1 Report
Comments and Suggestions for Authors
The paper has been improved, however some steps still need to be fixed, for example as the Authors refer to the 2016 meta-analysis by Park and colleagues (lines 138–139) the text is still confused.
A general re-evaluation of the paper could be useful.
Comments on the Quality of English LanguageThe English Language should be improved
Author Response
The paper has been improved, however some steps still need to be fixed, for example as the Authors refer to the 2016 meta-analysis by Park and colleagues (lines 138–139) the text is still confused.
A general re-evaluation of the paper could be useful.
We revised this section as requested and carefully read the full manuscript again for other linguistic errors/typos.